# The Impact of Demographic, Clinical Characteristics and the Various COVID-19 Variant Types on All-Cause Mortality: A Case-Series Retrospective Study

**DOI:** 10.3390/diseases10040100

**Published:** 2022-11-07

**Authors:** Faryal Khamis, Salah Al Awaidy, Muna Ba’Omar, Wessam Osman, Shabnam Chhetri, Zaiyana Ambusaid, Zakariya Al Fahdi, Jaber Al Lawati, Khalsa Al Sulaimi, Salma Ali Al Bulushi, Maher Al Bahrani, Ibrahim Al-Zakwani

**Affiliations:** 1Department of Infectious Disease, The Royal Hospital, Muscat, PC 111, Oman; 2Ministry of Health, Sultanate of Oman, Muscat, PC 100, Oman; 3Department of Medicine, Royal Hospital, Ministry of Health, Muscat, PC 111, Oman; 4Department of Medicine, Nizwa Hospital, Ministry of Health, Nizwa, PC 611, Oman; 5Department of Nursing, Royal Hospital, Ministry of Health, Muscat, PC 111, Oman; 6Department of Anaesthesia, Royal Hospital, Ministry of Health, Muscat, PC 111, Oman; 7Department of Pharmacology & Clinical Pharmacy, College of Medicine & Health Sciences, Sultan Qaboos University, Al Khoudh, PC 123, Oman

**Keywords:** COVID-19, epidemiological and clinical characteristics, outcomes, SARS-CoV-2, mortality, Oman

## Abstract

(1) Background: Severe acute respiratory syndrome coronavirus 2 (SARS-CoV-2) has rapidly evolved into a pandemic affecting virtually every country in the world. We evaluated the demographic, clinical, laboratory, and all-cause mortality of moderate and severe COVID-19 patients admitted to a tertiary care hospital in Oman during the different COVID-19 waves and variant types. (2) Methods: A case-series retrospective study was carried out between 12 March 2020 and 30 June 2022. All adults over the age of 18 with laboratory-confirmed COVID-19 were enrolled. Analyses were performed using univariate and multivariate statistics. (3) Results: A total of 1462 confirmed cases enrolled with the mean age of the cohort was 55 ± 17 years with significant differences among the groups (*p* = 0.006). A total of 63% and 80% of the patients were males and citizens of Oman, respectively. Patients infected with the Alpha COVID-19 variant type were more likely to have acute respiratory distress syndrome (ARDS) (*p* < 0.001), stay longer in the hospital (*p* < 0.001), and get admitted to the intensive care unit (ICU) (*p* < 0.001). At the same time, those who had the Omicron COVID-19 type were more likely to have renal impairment (*p* < 0.001) and less likely to be associated with non-invasive ventilation (NIV) (*p* = 0.001) compared with other COVID-19 variant types. The Delta (adjusted odds ratio (aOR), 1.8; 95% confidence interval (CI): 1.22–2.66; *p* = 0.003) and Omicron (aOR, 1.88; 95% CI: 1.09–3.22; *p* = 0.022) COVID-19 variant types were associated with higher all-cause mortality when compared to the initial COVID-19 variant. Old age (aOR, 1.05; 95% CI: 1.04–1.06; *p* < 0.001), the presence of respiratory disease (aOR, 1.58; 95% CI: 1.02–2.44; *p* = 0.04), ICU admission (aOR, 3.41; 95% CI: 2.16–5.39; *p* < 0.001), lower eGFR (aOR, 1.61; 95% CI: 1.17–2.23; *p* = 0.004), and ARDS (aOR, 5.75; 95% CI: 3.69–8.98; *p* < 0.001) were also associated with higher mortality while NIV requirements were associated with lower odds of dying (aOR, 0.65; 95% CI: 0.46–0.91; *p* = 0.012). (4) Conclusions: Alpha and Delta variants were associated with a longer hospital stay, need for intensive care, mechanical ventilation, and increased mortality. Old age, cardiac renal dysfunction were commonly associated with Omicron variants. Large-scale national studies to further assess the risk factors for mortality related to COVID-19 waves are warranted.

## 1. Introduction

More than two years into the coronavirus disease 2019 (COVID-19) pandemic, over 300 million cases and 6 million deaths have been reported around the world [1]. The pandemic is confronting the world with pressing and unresolved challenges. The current pandemic marks the third fatal coronavirus outbreak following severe acute respiratory syndrome in 2003 and Middle East respiratory syndrome in 2012 [2,3]. 

In Oman, the first two cases of COVID-19 were notified on 24 February 2020 and were related to travel to the Islamic Republic of Iran [4]. Oman has a population of around five million people, and as of 3 September 2022, the total number of laboratory-confirmed COVID-19 cases reached 397,993 with 4628 deaths. The COVID-19 vaccination campaign started among adults over the age of 18 years in December 2020, and by 26 August 2022, about 7,089,974 vaccine doses were administered, with 138.84 total doses administered per 100 people [5]. By the end of August 2022, the nationwide coverage rates for at least single and double doses of the COVID-19 vaccination were 97% and 90.7%, respectively. 

The country witnessed four COVID-19 waves (Figure 1). While a precise epidemiological definition of a wave remains controversial, the most agreeable description is a rising number of reported cases, followed by a drastic surge and a subsequent gradual falling-off [6]. The first wave of cases was due to the novel Wuhan strain as cases started to increase in mid-April 2020 and reached the peak in July 2020, during the time when people gathered for the Islamic festivals [7,8]. This was followed by two major peaks in October 2020 and June 2021 that were predominated by Alpha and Delta variants, respectively. The wide implementation of mandatory vaccination programs by June 2021 reduced the number of new infections. However, another peak surged in February 2022, and due to the waning immunity from vaccinations and ease of COVID-19 restrictions, the country witnessed a fourth wave of the pandemic, mainly driven by the Omicron variant [5]. 

Most patients with COVID-19 disease experience mild to moderate respiratory illness and recover without special medical treatment. However, several studies have demonstrated that the elderly and individuals suffering from chronic diseases such as diabetes, respiratory disease, heart disease, and cancer are more likely to develop severe disease [9,10] and abnormalities in various biomarkers. Clinical and epidemiological endpoints may be important in understanding the severity of the disease during the COVID-19 waves with its variants.

Due to a potential risk of increased transmission of the virus, the severity of the infected individuals, and the ability to evade antibodies from vaccines, the five severe acute respiratory syndrome coronavirus 2 (SARS-CoV-2) variants of Alpha (B.1.1.7), Beta (B.1.351), Gamma (P.1), Delta (B.1.617.2), and Omicron (BA.1 and BA.2) drew the greatest global attention [11]. 

As the pandemic evolved, numerous variants of SARS-CoV-2 emerged around the world. The World Health Organization (WHO) reported that the findings of the severity of the disease were the most uncertain in studies of phenotypic effects of SARS-CoV-2 variants of concern (VOCs). There is a paucity of published information on the severity of virus variant disease [12]. Due to the wide scale of clinical presentations, research on clinical and epidemiological factors among different SARS-CoV-2 variants which can predict prognosis is of paramount importance [13]. Existing global reports show variations in the characteristics of the different waves [14,15,16,17]. Several factors could have impacted the surge and decline of the reported cases, including the effectiveness of vaccination, public health interventions, mitigation strategies, human behavior, infection prevention measures, virus mutations, and prevalence of anti-SARS-CoV-2 antibodies among the population [18,19,20,21]. 

In this study, we describe epidemiological characteristics, clinical and laboratory features, and mortality outcomes of patients with moderate and severe COVID-19 illness admitted to a tertiary care hospital in Oman during the various COVID-19 variant types (Wuhan, Alpha, Delta, and Omicron).

## 2. Methods

### 2.1. Study Design and Data Collection

We conducted a retrospective study that enrolled patients from 12 March 2020 to 30 June 2022. The inclusion criteria were as follows: (1) patients above 12 years of age with laboratory-confirmed COVID-19 by viral nucleic acid detection using RT-PCR with samples from the nasopharynx and respiratory secretions; (2) patients with a clinical spectrum of moderate, severe, and critical COVID-19 disease based on U.S. Department of Health & Human Services, National Institutes of Health (NIH) criteria (www.covid19treatmentguidelines.nih.gov accessed on 24 September 2022) who required hospitalization; (3) patients who underwent complete laboratory tests on admission (routine blood tests, biochemistry analysis, liver functions test, CRP, LDH) and clinical recording at admission; and (4) patients who underwent chest imaging on admission. We excluded children below 12 years of age; any patient who had not done SARS-CoV-2 PCR, was not hospitalized, and was admitted after the study censored date. The data were censored at the time of the data cutoff, which occurred on 30 June 2022. Patients hospitalized after the censored date were not included. 

Data for this study were obtained from the Royal Hospital (RH) COVID-19 Registry. The Registry is a record of allhospitalized patients with confirmed SARS-CoV-2-infection and is based on data extracted from the hospital electronic medical records (EMRs). Data collectors were trained to ensure the completeness and accuracy of the data collected. Patient data were documented on standard reporting forms and transferred in the form of an electronic Microsoft Excel spreadsheet software program. Duplicate entries were omitted. Data extracted include the baseline demographic characteristics (gender, age, occupation, place of residency, and nationality), information on the date of onset of illness, date of COVID-19 laboratory confirmation, COVID-19 vaccination, risk factors, and underlying co-morbidities, clinical symptoms, and signs on presentation, need for oxygen support, laboratory parameters, radiological features, drug therapy, duration of hospitalization and patients’ outcomes (Intensive care unit (ICU) transfer, length of stay or mortality). Patients were followed throughout the hospitalization and up until discharge or death.

Cases were diagnosed using national definitions of suspected and confirmed COVID-19 cases [22]. Confirmed COVID-19 cases were defined as: patients with suspected COVID-19 that tested positive for SARS-CoV-2 as per testing standards, irrespective of clinical signs and symptoms. WHO classification of COVID-19 clinical severity was used where COVID-19 cases were categorized into asymptomatic, mild, moderate, severe, or critical disease. Mild disease includes “symptomatic patients meeting the case definition for COVID-19 without evidence of viral pneumonia or hypoxia. Moderate disease is defined as pneumonia without age-specific signs and symptoms of severe pneumonia. Severe disease refers to pneumonia cases with an age-specific clinical picture of acute respiratory distress syndrome (ARDS) [23]. 

All patients received the standard of care as per the MOH Protocol for Management of Hospitalized Adult Patients with COVID-19 Infection [MoH/DGSMC/PRT/003/Vers.2, Ministry of Health, Oman] and ICU Protocol for Management of COVID-19 [MoH/UHAO/PRT/002/Vers.01]. 

### 2.2. Laboratory Procedures

Samples were analyzed using the WHO and national guidelines. The nasopharyngeal swabs were placed into a collection tube with 150 μL of virus preservation solution, and total RNA was extracted within 2 h by Liferiver Novel Coronavirus (2019-nCoV) Real Time Multiplex RT-PCR Kit [Shanghai Zhijiang Biotechnology Co., Ltd. (ZJ Bio-Tech), Shanghai, China]. This was the same kit that was used for the qualitative detection of a novel coronavirus, which was identified in 2019 in Wuhan City, China, by real-time PCR systems. The assays included a positive and an internal control. The probes specific for SARS-CoV-2 RNA were labeled with the fluorophore FAM (ORF1ab), HEX/VIC/JOE (gene N), and Cal Red 610/ROX/TEXAS RED (gene E). The probe specific for Internal Control (IC) is labeled with the fluorophore Cy5. Real-time PCR was performed upon an Applied BiosystemsTM 7500 Fast Real-Time PCR System following the cycling and fluorescence acquisition parameters detailed in the LifeRiver Novel Coronavirus (2019-nCoV) Real Time Multiplex (RR-0479-02). As per manufacturer recommendations, the samples were tested from bronchoalveolar lavage, sputum, swab, and endotracheal aspirate. Samples and controls were assigned a cycle threshold value. The samples were interpreted as either “COVID-19 Positive”, “COVID-19 Negative”, “Potential positive”, or “result invalid”. 

### 2.3. Ethical Approval

The research has been approved by Royal Hospital Research and Ethics Committee (SRC#26/2020). Patients names and their corresponding hospital medical record numbers were coded and remained confidential. 

### 2.4. Statistical Analysis

Descriptive statistics were used to describe the data. For categorical variables, frequencies and percentages were reported. Differences between groups were analyzed using Pearson’s χ^2^ tests (or Fisher’s exact tests for expected cells < 5). The continuous variables, age, mean and standard deviation, were used to summarize the data and differences between groups which were analyzed using ordinary least squares (OLS) regression. The discrete variables, length of hospital stay, median, and interquartile range, were used to present the data and differences analyzed using the Kruskal–Wallis test. The impact of the various COVID-19 variant types (Alpha, Delta, Wuhan, Omicron), as well as demographic and clinical characteristics on all-cause mortality, were analyzed using multivariable logistic regression utilizing the simultaneous method. An a priori two-tailed level of significance was set at 0.05. Statistical analyses were conducted using STATA version 16.1 (STATA Corporation, College Station, TX, USA). 

## 3. Results

A total of 1462 confirmed cases of COVID-19 have been included. Demographic and clinical characteristics are outlined in Table 1. The overall mean age of the cohort was 55 ± 17 years, with significant differences among the groups (*p* = 0.006). A total of 63% (n = 924) and 80% (n = 1130) of the patients were males and Oman citizens, respectively. The three most prevalent comorbidities were hypertension (51%; n = 746), diabetes mellitus (47%; n = 681), and heart diseases (20%; 288/1409), with statistically significant differences among the groups in regard to diabetes mellitus (*p* = 0.007) and heart diseases (*p* < 0.001). Those that were infected with the Alpha COVID-19 variant type were more likely to have acute respiratory distress syndrome (ARDS) (*p* < 0.001), stay longer in hospital (11 days *p* < 0.001), and get admitted to the ICU (*p* < 0.001) while those that had the Omicron COVID-19 type were more likely to have renal impairment (*p* < 0.001) and less likely to be associated with non-invasive ventilation (*p* = 0.001) compared with other COVID-19 variant types.

In adjusting for other factors in the multivariate logistic model in Figure 2, the Delta (adjusted odds ratio (aOR) 1.8; 95% CI: 1.22–2.66; *p* = 0.003) and Omicron (aOR 1.88; 95% CI: 1.09–3.22; *p* = 0.022) COVID-19 variant types were associated with higher all-cause mortality when compared to the Wuhan COVID-19 variant. Additionally, old age (aOR 1.05; 95% CI: 1.04–1.06; *p* < 0.001), the presence of respiratory disease (aOR 1.58; 95% CI: 1.02–2.44; *p* = 0.04), ICU admission (aOR 3.41; 95% CI: 2.16–5.39; *p* < 0.001), lower Glomerular filtration rate (ecGFR) (aOR 1.61; 95% CI: 1.17–2.23; *p* = 0.004), and ARDS (aOR 5.75; 95% CI: 3.69–8.98; *p* < 0.001) were also associated with higher mortality while non-invasive mechanical ventilation (NIV) requirements were associated with lower odds of dying (aOR 0.65; 95% CI: 0.46–0.91; *p* = 0.012). In this cohort, 48% (707/1461) of patients required ICU admission, and 393 (27%) succumbed to the disease.

## 4. Discussion

We conducted a retrospective case series study of 1462 patients with laboratory-confirmed COVID-19 hospitalized from March 2020 to June 2022. In this cohort, about 48% of patients required care in the ICU, and 27% died.

Several studies have compared the characteristics of hospitalized patients during different waves of COVID-19 [15,24,25,26,27]. However, these studies were early in the pandemic and reported the first two waves using different variables and parameters. To the best of our knowledge, only a few studies have performed a comparison of the clinical characteristics and outcomes of the four waves [28,29,30].

In a study from the Democratic Republic of the Congo, there were no significant differences in the demographics across the four waves, the sex ratio remained constant, but the median age at death was high (above 60 years), even though it was similar across all four waves. Delta variant dominated the third wave causing a high fatality rate, while the Omicron variant during the fourth wave was less virulent and less fatal [27,28]. Similarly, a comparison of four epidemic waves of COVID-19 in Malawi aimed to explore the relationship between viral lineage and patient outcome suggested that patients with severe COVID-19 disease were more likely to die during the Delta wave [28,29]. In another study from Western Cape Province, South Africa, that included 5144 patients from wave four and 11,609 from prior waves, the risk of all outcomes was lower in wave four compared to the third wave driven by Delta wave three, although the risk reduction was less obvious when adjusted for prior vaccination and/or confirmed infections [29,30].

In this study, we found a significant difference in the clinical characteristics and outcomes among hospitalized patients during the four waves of the COVID-19 pandemic in Oman. Our results indicate that the second and third waves were more serious than the other waves. This correlates with the increase in the severity of disease and community transmission that was observed with Alpha and Delta variants, respectively [31,32,33,34].

Several studies suggested that the COVID-19 surges appeared as a result of mutations in the SARS-CoV-2 virus and the emergence of new variants that are more virulent and transmissible [35,36,37,38,39]. The occurrence of mutations in the SARS-CoV-2 viral genome during replication is a natural phenomenon that has led to a number of novel variants [34,35]. The virus evolves at a rate of approximately 1.1 × 10^3^ substitutions per site per year, approximately one substitution every 11 days, but only a few mutations are deemed to be significant [39,40]. SARS-CoV-2 is an efficient virus that dysregulates innate and adaptive immune responses and has the ability to adapt to other hosts [40,41,42]. Thus, continuous monitoring and surveillance for the detection of new variants are crucial.

In Oman, a molecular study on the whole-genome sequence was conducted by Al-Mahruqi et al. early in the pandemic (March to May 2020) from a cluster of 94 cases. In using both whole viral genome sequencing and epidemiological surveillance data, the most prevalent mutation was P323L (94.7%) in the non-structural protein, followed by the D614G (92.6%) in the Spike glycoprotein. The D614G mutation of B1 lineages (GR, G, and GH clades) was also reported from 116 countries. Furthermore, two unique mutations I280V and R502C, were also detected at a lower frequency (5.3%) and 2 (2.1%), respectively [42,43]. Subsequently, all the other variants of concern were detected in the country [MOH press communication].

In this cohort, about 98% of the infected hospitalized patients during the Omicron wave were nationals, compared to 77% during the first wave caused by the novel COVID-19 virus. Ethnicity has not been confirmed as a risk factor for negative outcomes in COVID-19 patients [24,25,26,27,28,29,30,31,32,33,34,35,36,37,38,39,40,41,42,43,44,45]. The potential factors behind the differences in the first wave are the dense living conditions of the non-nationals and their lack to prompt access to medical care. Additionally, it might be possible that non-nationals had less vaccine hesitancy than nationals, particularly for the booster doses of COVID-19 vaccination. With a total population of the country reaching 5,385,432, of which the non-national population is 1,553,981, as of 20 February 2022, more than 30% of non-nationals (448,825) and 10% of nationals (153,401) received a booster dose of the vaccines [Press communication, MOH].

In terms of age, there were significant differences across the waves of the COVID-19 pandemic, with older patients more likely to be affected by the Omicron variant than the younger age groups. This is probably due to evading of the antibodies that was observed in the older population after COVID-19 vaccination. The association between age and the severity of the disease has been reported in several other studies [45,46,47,48,49,50,51]. In a nationwide cohort covering 37% of all COVID-19 cases in England, Omicron cases had a 59% lower risk of hospital admission, a 44% lower risk of any hospital attendance, and a 69% lower risk of death than that of confirmed delta cases. There was found strong evidence of age dependence in the magnitude of this risk reduction in severity [52].

The most prominent co-morbidity in the first three waves was diabetes mellitus, while heart diseases were more common among patients hospitalized during the Omicron wave. The majority of the patients were asymptomatic and were detected during screening for procedures or in accordance with the infection control policy in our hospital. Similarly, renal impairment was more prevalent in patients hospitalized with COVID-19 during the Omicron wave. This could be attributed to our local policy, as screening for COVID-19 in dialysis and cardiac wards was mandatory. In a recent meta-analysis of 49,048 patients from 142 studies, the overall incidence of acute renal impairment in patients admitted with COVID-19 was 28.6% [53]. However, the rate of acute renal impairment related to COVID-19 has been lower in the subsequent waves [54,55]. Contrary to our findings, no significant differences in sociodemographic, lifestyle information, or laboratory data were observed in a study from Denmark that compared the demographic of those admitted during the first and second waves of COVID-19 [56]. Nevertheless, in many studies, co-morbidities were found to be significant risk factors for critical COVID-19 disease in all the waves [56,57].

In the current study, patients had a longer length of hospital stay (LOS) during the first three waves, with a median of 11 days, while the median LOS during the fourth Omicron wave was 5 days. Hospitalized patients with the Alpha variant were more likely to present with shortness of breath, ARDS, and tend to have worse clinical outcomes, including the need for ICU care. Meanwhile, higher mortality rates were observed in patients admitted with the Alpha and Delta variants. On the contrary, patients with the Omicron variant had a milder course of illness, less need for oxygen supplement or mechanical ventilation, and significantly shorter LOS compared with the other COVID-19 variants. Similarly, in a study from South Africa conducted by Maslo et al., the proportion of patients requiring oxygen therapy significantly decreased from 74% in wave three to 17.6% in wave four, as did the percentage receiving mechanical ventilation. Furthermore, admission to ICU decreased from 29.9% in wave three to 18.5% in wave four. The median LOS of 7 to 8 days in previous waves decreased to 3 days in wave four [58].

A number of studies investigated the risk factors associated with mortality in patients with COVID-19 [59,60,61]. Studies from Denmark, Japan, Iran, and Spain have reported a milder course of the disease and lower mortality among hospitalized patients with COVID-19 during the later waves [15,56,62,63]. In this cohort, we identified the most significant risk factors associated with COVID-19 deaths in patients from all the waves. Older age, previous infection with Delta variant, prior history of respiratory disease, presentation with ARDS, renal impairment, and the need for ICU care were identified as risk factors for increased mortality. Furthermore, on multivariant analysis, mortality was higher in patients with the Omicron variant compared to the Wuhan COVID-19 variant, although most of the patients had asymptomatic or mild COVID-19 illness detected by routine screening. The higher death rate could be attributed to the associated co-morbidities such as cardiac diseases, renal impairment, or rarely the delay in presentation to receive the needed medical care. There is substantial evidence that infection with the Omicron variant causes less severe disease compared to the Delta variant; however, Omicron infections can still be severe, particularly in older age, individuals with co-morbidities, and unvaccinated individuals [64,65,66].

There are several reasons for the differences in the course of illness between the COVID-19 waves, such as the susceptibility of the older and vulnerable population to acquire COVID-19 infection early in the pandemic, wide implementation of restriction policies, and social distancing, scale-up of screening and diagnostic testing, availability of vaccinations and improved patients’ outcomes with effective and timely clinical management strategies and therapeutics. Finally, the level of global preparedness and accumulative experiences that were gained over time were major decisive factors in the aforementioned outcomes [67,68,69,70]. Nevertheless, there is still a tremendous need to develop newer, sensitive, and rapid diagnostic tests, as well as novel vaccines and pharmaceutical therapies to combat the new variants of concern and the ongoing risk.

Although our study provided a sufficient sample size, it has several noteworthy limitations. First is the retrospective nature. Second, the study focused on hospitalized patients in a single center with moderate to severe COVID-19 infection with potential bias in estimating the population risk factors. Third was constraints in the analysis of data as some laboratory investigations and diagnostic workup were missing. Fourth was the calculation of mortality differences among hospitalized patients can be confounded by multiple variables such as other comorbidities, early access to care, therapeutics used, and type of vaccines received. Finally, the study could not assess the effect of using antivirals or immunomodulatory agents on the outcome. Therefore, the results need to be extrapolated with vigilance.

## 5. Conclusions

We found a significant difference in the clinical characteristics and outcomes among hospitalized patients during the four waves of the COVID-19 pandemic in Oman. Risk factors such as diabetes mellitus were prominent in the first three waves, while heart diseases and renal dysfunction were more prevalent among patients with the Omicron variant. Furthermore, LOS need for mechanical ventilation and mortality rates were lower for the Omicron variant than patients admitted with the Alpha and Delta variants. Future studies evaluating risk variables for critical disease in multiple COVID-19 waves would be of valuable significance to medical literature.

## Figures and Tables

**Figure 1 diseases-10-00100-f001:**
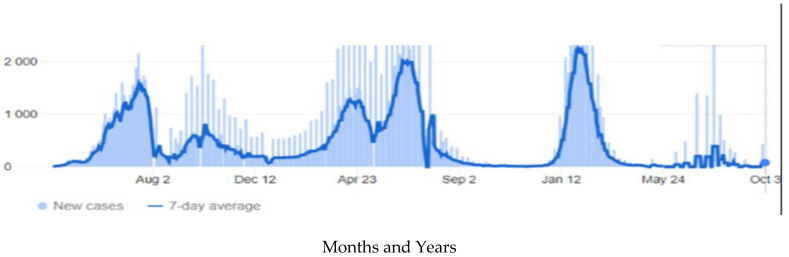
COVID-19 waves, 2020–2022, Oman.

**Figure 2 diseases-10-00100-f002:**
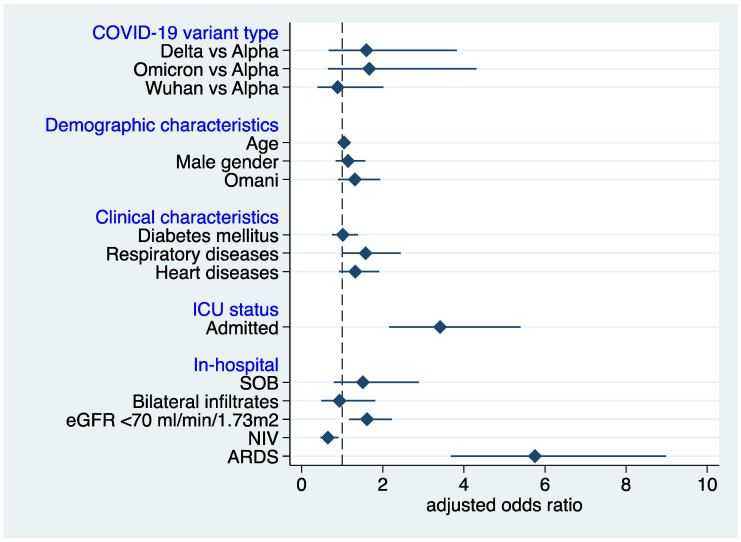
Adjusted odds ratios of the predictors of mortality of admitted COVID-19 patients (N = 1462). ICU, intensive care unit; SOB, shortness of breath; eGFR, estimated glomerular filtration rate; NIV, non-invasive ventilation; ARDS, acute respiratory distress syndrome.

**Table 1 diseases-10-00100-t001:** Demographic, clinical characteristics, and outcomes stratified by COVID-19 variant type.

Characteristic, n (%) Unless Specified Otherwise	All (N = 1462)	COVID-19 Variant				*p*-Value
		Wuhan (n = 988)	Alpha (n = 40)	Delta (n = 209)	Omicron (n = 225)	
Demographic						
Age, mean ± SD, years	55 ± 17	54 ± 16	57 ± 16	57 ± 17	58 ± 20	0.006
Male	924 (63%)	637 (64%)	24 (60%)	129 (62%)	134 (60%)	0.504
Oman citizen (N = 1409)	1130 (80%)	764 (77%)	31 (78%)	167 (80%)	168 (98%)	<0.001
Comorbidities						
Diabetes mellitus	681 (47%)	488 (49%)	21 (53%)	81 (39%)	91 (40%)	0.007
Hypertension	746 (51%)	505 (51%)	24 (60%)	94 (45%)	123 (55%)	0.136
Respiratory disease *	162 (11%)	113 (11%)	9 (23%)	21 (10%)	19 (8.4%)	0.065
Heart disease (N = 1409) **	288 (20%)	182 (18%)	7 (18%)	35 (17%)	64 (37%)	<0.001
Liver disease	55 (3.8%)	43 (4.4%)	1 (2.5%)	3 (1.4%)	8 (3.6%)	0.214
Intensive care unit (ICU) admission (N = 1461)	707 (48%)	509 (52%)	27 (68%)	117 (56%)	54 (24%)	<0.001
Bilateral infiltrate (N = 1461)	1099 (75%)	834 (84%)	33 (83%)	161 (77%)	71 (32%)	<0.001
Length of hospital stay (N = 1454)	8 (4–15)	8 (4–15)	11 (6–23)	9 (5–18)	5 (3–10)	<0.001
Kidney impairment (N = 1453)	523 (36%)	305 (31%)	16 (40%)	81 (39%)	121 (56%)	<0.001
Non-invasive ventilation	403 (28%)	275 (28%)	10 (25%)	75 (36%)	43 (19%)	0.001
Acute respiratory distress syndrome (ARDS) (N = 1457)	573 (39%)	417 (42%)	24 (60%)	95 (45%)	37 (17%)	<0.001
In-hospital mortality	393 (27%)	250 (25%)	15 (38%)	76 (36%)	52 (23%)	0.002

* Included conditions such as chronic obstructive pulmonary diseases, asthma, interstitial lung diseases, bronchiectasis, and tuberculosis. ** Included conditions such as coronary artery diseases, valvular heart diseases, congestive cardiac failure, and arrhythmias.

## Data Availability

The data sets used and/or analysed during the current study are available from the corresponding author upon reasonable request.

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
