# Peer review of "The Impact of Demographic, Clinical Characteristics and the Various COVID-19 Variant Types on All-Cause Mortality: A Case-Series Retrospective Study"

_diseases, 2022, doi:10.3390/diseases10040100_

Round 1

Reviewer 1 Report

Khamis’22 input-

1.      Line 123- Briefly mention platform used to conduct RT-PCR and assay parameters used to conduct the reactions.

2.      Line 155- Table 1- it would be neutral to use male: female or female: male ratio, than male gender in second row.

3.      Line 155- Table 1- under “Comorbidities” there are two rows, one as “Respiratory disease”, other as “Shortness of breath”.  Respiratory disease is too vague comorbidity, it needs to be mentioned which respiratory disease is under consideration. If multiple respiratory diseases are grouped together then one option would be to label them as “Respiratory diseases” If so , in foot note, authors have to mention which respiratory diseases were included such as COPD (Chronic obstructive pulmonary disease), TB (tuberculosis). 

Shortness of breath is symptom of array of respiratory and cardiac diseases.  This cannot be considered as comorbidity.  If shortness if breath is from respiratory elements, it should be covered in “Respiratory diseases” category.

4.      Line 165- 166- re-write the sentence. Instead of “died”, “succumbed to the disease” can be used.

5.      Result section- instead of putting two tables, data can be presented in more appealing format such as pie charts, columns.

6.      Line 215-216- what is the purpose of highlighting part of the sentence.

7.      Line 167 – 276- Discussion section has to be re-organized and re-written with proper structure and flow.  

Reviewer 2 Report

The introduction and methodology are correct, but some variables looks too inaccurate, for instance the heart pathologies can include different patterns of cardiac alterations. In the discussion the authors could be more critical about the limitations of your work. The bibliographic revision is complete.   

Reviewer 3 Report

The authors present a statistical analysis of COVID cases in Oman.  The study is of relevance, and the large sample size make the study appealing to a more general epidemiological and infectious disease audience.  Overall, the authors have a laid a decent foundation for their analysis. However, the article is missing some analyses that are pivotal, and there are some methodical clarifications must be addressed.

MAJOR

1.  The authors study design and data collection section (Section 2.1) is too vague.  There needs to be  paragraph detailing the inclusion and exclusion criteria.  How many people were initially enrolled?  How much temporal follow-up or metrics were required to be retained in the present study's results?  For example, if someway enrolled on June 29, 2022 (the day before the stated study end date), were they still included? How many subjects were excluded due to censoring or missing data?  If there are patients included with missing data (which is almost always the case with these types of studies), please provide sample size and degree of missingness for each variable as an appendix table.  Additionally, the degree of censored data should be included where patients were not followed to the final end point.

2. For 2-tailed pairwise tests, it is not stated if correction factors for multiple comparisons were employed (example: Bonferroni, Turkey, etc.). Any time there are multiple comparisons for the same group, there should be a statistical correction factor for multiple comparison for the analysis to be technically correct; otherwise, Type I errors are likely. This needs to be clearly stated with a description of the correction factor(s) utilized. If using a generic setting in STATA that has the correction factor, the setting should be stated inside the Methods.

3. The authors state that they perform logistical regression. That would make sense to look at overall relationship mappings. However, given the likely censoring of data and the fact that survival was a key outcome, was there a reason why Cox regression was not chosen to specifically examine hazard ratios and CI of variables on patient survival?  This a more common method for mapping the impact of variables on survival.  

4. Typically odds ratios are also presented in a forest plot.  Given the extent of tabular data (which should remain), a forest plot visualization would greatly enhance the ease of interpretability for fast reader evaluation of the key results.

5. No where do the authors mention what the "standard of care" treatments were for these hospitalized patients.  Did they all have similar treatment plans?  What were they? Some basics should be provided such as: was there a standard IV hydration protocol; were patients given antibacterials to prevent secondary bacterial infection; other repurposed drugs or supplements meant to assist with COVID-19, such as steroids for respiratory inflammation or nebulized albuterol, etc. What were the general criteria for mechanical ventilation, NIV, and ICU admission, etc.? Obviously there will be some differences in treatment on a per-patient basis. Nonetheless, it its pivotal to understand some of this center's standard of care protocols for COVID-19 patients.  Without this context, it's difficult to compare ICU admissions, hospital duration, and survival to other COVID-19 studies in other locations.  And the ability to have context for comparison is critical for researchers in the field.

6. In the Discussion, the authors state: "To the best of our knowledge, only a few studies have performed a comprehensive comparison of the 201 clinical characteristics and outcomes of the 4 waves."  However, no citations are given after this sentence for these studies.  Both references and summaries of those studies is necessary to compare the author's study is critical material for the Discussion. Please cite the studies to which you refer and summarize your major findings compared to their findings.

MINOR:

1. Table 1 says "shortness of breadth" but breadth should be "breath".

2. Table 2 does not have a table or table caption.

3. The descriptive statistics for continuous variables should also be provided in violin plots, namely age and length of hospital stay. The authors do not clearly state throughout what the error bars are denoting (standard deviation, interquartile range, etc.). This should be stated plainly in each figure/table caption as well.

4. There is extensive overuse of the first-person "we" which is not standard in such clinical articles.  

5. Figure 1 resolution is poor.

6. The Conclusion paragraph (Section 5) and conclusion sentence in the abstract is very vague and provides no real value.  Please revise Conclusions section to highlight major findings and provide meaningful context. Please correspondingly revise the Conclusion sentence in the abstract.

7. Line 229 "In contrary....". This should be "in contrast". There are several places where English is poor and should reviewed by a native English speaker.

8. In line 229-231 of the Discussion, the authors state they did not find any socioeconomic or lifestyle differences in their study unlike the cited Denmark study. However, no results are shown in the authors' present study with those variables. If this is going to be mentioned in the Discussion, the authors need to provide the analysis in the results and corresponding data in the descriptive statistics table or in an appendix table.   This reviewer believe this result/finding would be of interest to readers and, thus ,should be included in the main results.

Round 2

Reviewer 1 Report

Thanks for doing changes to the necessary level.

Author Response

No comments made by the reviewers

Reviewer 3 Report

The authors have successfully revised the manuscript.

Author Response

No further comments made by the reviewers.
